# Time Expression Analysis and Recognition Using Syntactic Types and Simple Heuristic Rules

## Abstract

Extracting time expressions from free text is a fundamental task for many applications. We analyze the time expressions from four datasets and observe that only a small group of words are used to express time information, and the words in time expressions demonstrate similar syntactic behaviour. Based on the observations, we propose a type-based approach, named SynTime, to recognize time expressions. Specifically, we define three main **syntactic types**, namely *time token*, *modifier*, and *numeral*, to group time-related regular expressions over tokens. On the types we design **simple heuristic rules** to recognize time expressions. In recognition, SynTime first identifies the time tokens from raw text, then searches their surroundings for modifiers and numerals to form time segments, and finally merges the time segments to time expressions. As a light-weight rule-based tagger, SynTime runs in real time, and can be easily expanded by simply adding keywords for the text of different types and of different domains. Experiment results show that SynTime outperforms state-of-the-art methods on benchmark datasets and tweets data.

## 1 Introduction

Time expression plays an important role in information retrieval and many applications in natural language processing (Alonso et al., 2011). Recognizing time expressions from free text has attracted considerable attention since last decade (Verhagen et al., 2007, 2010; UzZaman et al., 2013).

Time expression recognition main involves two kinds of methods, rule-based method (Strotgen and Gertz, 2013; Strötgen and Gertz, 2010; Chang and Manning, 2012) and machine learning based method (Bethard, 2013; Lee et al., 2014). Rule-based time expression taggers could recognize most time expressions with carefully designed rules, while they could not recognize the time expressions that are not matched in any explicit rule. Machine learning based methods require training data for good performance, and may not recognize less frequent time expressions.

In our study, we analyze the time expressions in four datasets: TimeBank (Pustejovsky et al., 2003b), Gigaword (Parker et al., 2011), Wiki-Wars (Mazur and Dale, 2010), and Tweets. From the analysis, we make four **observations**. First, most time expressions are very short, with 80% of time expressions containing no more than three tokens. Second, the vocabulary used to express time information is very small, with a small group of keywords. Third, at least 93% of time expressions contain at least one time token. The last observation is that words in time expressions demonstrate similar syntactic behaviors. All the observations relate to **the principle of least effort** (Zipf, 1949). That is, people will act under the least effort in order to minimize the cost of energy at both individual level and collective level to language usage (Zipf, 1949). Time expression is part of language and acts as an interface of communication. Short expressions, small vocabulary, occurrence, and similar syntactic behaviors all reduce the cost of energy required to communicate.

Based on the observations, we propose a type-based approach, named SynTime ('Syn' is from Syntactic), to recognize time expressions. Specifically, we define 3 main types, namely *time token*, *modifier*, and *numeral*, to group time-related regular expressions over tokens. Time tokens are the words that explicitly express time information,

such as time units (*e.g.,* 'month'). Modifiers modify time tokens; they may appear before or after time tokens, *e.g.,* 'several' and 'ago' in 'several months ago.' Numeral are ordinals and numbers. From free text SynTime first identifies time tokens, and then recognizes modifiers and numerals.

Naturally, SynTime is a rule-based tagger. The key difference between SynTime and other rule-based taggers lies in the way of defining **types** and the way of designing **rules**. The type definition of SynTime is inspired by part-of-speech in which "linguists group some words of language into classes (sets) which show similar syntactic behaviour." (Manning and Schutze, 1999) SynTime defines types for tokens according to their **syntactic** behaviors. Other rule-based taggers define types for tokens based on their *semantic* meaning. For example, SUTime defines 5 semantic modifier types, such as frequency modifiers; [1] while SynTime defines 5 syntactic modifier types, such as modifiers that appear before time tokens. (see Section 4.1 for details.) Accordingly, other rule-based taggers design rules for each type based on their meanings and deal with each type separately. SynTime designs rules based on the token types and their relative positions in time expressions. That is why we call SynTime a type-based approach. More importantly, other rule-based taggers design rules in a *fixed* way, including fixed length and fixed position. In contrast, SynTime designs rules in a **heuristic** method based on the idea of **boundary expansion**. The heuristic rules are quite **simple** that it makes SynTime much more flexible, expansible, and extremely light-weight, leading SynTime to run in real time.

We evaluate SynTime against three state-of-the-art baselines, namely, HeidelTime, SUTime, and UWTime, on three datasets, namely, TimeBank, WikiWars, and Tweets. TimeBank and WikiWars are benchmark datasets for time expression extraction. [2] Experiment results show that SynTime significantly outperforms the three state-of-the-art methods on TimeBank and Tweets datasets. On WikiWars, SynTime achieves comparable results. More importantly, SynTime achieves the best recalls on all three datasets and exceptionally good results on Tweets dataset. To sum up, we make the following contributions.

---

[1] https://github.com/stanfordnlp/CoreNLP/tree/master/src/edu/stanford/nlp/time/rules

[2] Gigaword dataset was not used in our experiments because the labels in the dataset was automatically generated by other taggers which may not be the ground truth labels.

- We analyze the time expressions from four datasets and make four observations. The observations provide evidence in terms of time expression for the principle of least effort (Zipf, 1949). We design SynTime based on the observations.

- We propose a type-based time expression tagger, SynTime, that defines syntactic types and designs simple heuristic rules for time expression recognition. SynTime provides an idea to simplify rule-based time tagger.

- We conduct experiments on three datasets, and the results demonstrate the effectiveness of SynTime against state-of-the-art baselines.

## 2 Related Work

Many research on time expression identification are reported in TempEval exercises (Verhagen et al., 2007, 2010; UzZaman et al., 2013). The task is divided into two subtasks: Recognition and normalization.

**Rule-based Time Expression Recognition.** Rule-based time taggers like GUTime, HeidelTime, and SUTime, predefine time-related words and rules (Verhagen et al., 2005; Strötgen and Gertz, 2010; Chang and Manning, 2012). HeidelTime (Strötgen and Gertz, 2010) hand-crafts rules with time resources like weekdays and months, and leverages language clues like part-of-speech to identify time expression. SUTime (Chang and Manning, 2012) designs fixed rules using a cascade finite automata (Hobbs et al., 1997) on regular expressions over tokens (Chang and Manning, 2014). It first identifies individual words, then expands them to chunks, and finally to time expressions. Rule-based taggers achieve very good results in TempEval exercises.

SynTime is also a rule-based tagger while the key differences between SynTime and other rule-based taggers are the way of defining types and of designing rules. SynTime defines types for tokens according to their syntactic behaviors and designs rules in a heuristic way.

**Machine Learning based Method.** Machine learning based methods extract features from the text and apply statistical models on the features for recognizing time expressions. Example features include character features, word features, syntactic features, semantic features, and gazetteer

features (Llorens et al., 2010; Filannino et al., 2013; Bethard, 2013). The statistical models include Markov Logic Network, Logistic Regression, Support Vector Machines, Maximum Entropy, and Conditional Random Fields (Llorens et al., 2010; UzZaman and Allen, 2010; Filannino et al., 2013; Bethard, 2013). Some models obtain good performance, and even achieve the highest $F_1$ of 82.71% on strict match in TempEval-3 (Bethard, 2013).

Outside TempEval exercises, Angeli *et al.* leverage compositional grammar and employ a EM-style approach to learn a latent parser for time expression recognition (Angeli et al., 2012). In the method named UWTime, Lee *et al.* handcraft a Combinatory Categorial Grammar (CCG) (Steedman, 1996) to define a set of lexicon with rules and use L1-regularization to learn linguistic context (Lee et al., 2014). The two methods explicitly use linguistic information. Particulaly in (Lee et al., 2014), CCG could capture rich structure information of language, similar to the rule-based methods. Tabassum *et al.* focus on resolving the dates in tweets, and use distant supervision to recognize time expressions (Tabassum et al., 2016). They use five time types and assign one of them to each word, which is similar to SynTime in the way of defining types over tokens. However, they focus only on the type of date, while SynTime can recognize all the time expressions and does not involve learning and runs in real time.

**Time Expression Normalization.** Methods in TempEval exercises design rules for time expression normalization (Verhagen et al., 2005; Strötgen and Gertz, 2010; Llorens et al., 2010; UzZaman and Allen, 2010; Filannino et al., 2013; Bethard, 2013). Because the rule systems has high similarity, Llorens *et al.* suggest to construct a large knowledge base as a public resource for the task (Llorens et al., 2012). Some researchers treat the normalization process as a learning task and use machine learning methods (Lee et al., 2014; Tabassum et al., 2016). Lee *et al.* (Lee et al., 2014) use AdaGrad algorithm (Duchi et al., 2011) and Tabassum *et al.* (Tabassum et al., 2016) use a log-linear algorithm to normalize the time expressions.

SynTime focuses only on time expression recognition, and the normalization could be achieved by using methods similar to the existing rule systems, because they are highly similar (Llorens et al., 2012).

Table 1: Statistics of the datasets. (a tweet here is a document.)

| Dataset | #Docs | #Words | #TIMEX |
|---------|-------|--------|--------|
| TimeBank | 183 | 61,418 | 1,243 |
| Gigaword | 2,452 | 666,309 | 12,739 |
| WikiWars | 22 | 119,468 | 2,671 |
| Tweets | 942 | 18,199 | 1,127 |

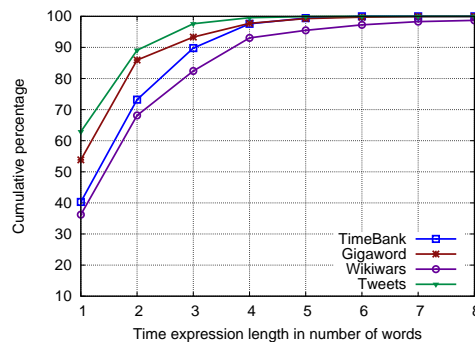

Figure 1: Length distribution of time expressions.

## 3 Time Expression Analysis

### 3.1 Dataset

We conduct analysis on four datasets: Time-Bank, Gigaword, WikiWars, and Tweets. Time-Bank (Pustejovsky et al., 2003b) is a benchmark dataset in TempEval series (Verhagen et al., 2007, 2010; UzZaman et al., 2013), consisting of 183 news articles. Gigaword (Parker et al., 2011) is a large *automatically* labeled dataset with 2,452 news articles and is used in TempEval-3. Wiki-Wars dataset is derived from Wikipedia articles about wars (Mazur and Dale, 2010). Tweets is our manually annotated dataset with 942 tweets of which each contains at least one time expression. Table 1 summarizes the datasets.

### 3.2 Observation

From the four datasets, we analyze their time expressions and make four observations.

**Observation 1** *Time expressions are very short. More than 80% of time expressions contain no more than three words and more than 90% contain no more than four words.*

Figure 1 plots the length distribution of time expressions. Although the texts are collected from different sources (*i.e.,* news articles, Wikipedia articles, and tweets) and vary in sizes, the length of time expressions follow a similar distribution. In particular, the one-word time expressions range

Table 2: The percentage of time expressions that contain at least one time token, and the average length of time expressions.

| Dataset | Percent | Average Length |
|---------|---------|----------------|
| TimeBank | 95.15 | 2.00 |
| Gigaword | 97.69 | 1.70 |
| WikiWars | 93.18 | 2.38 |
| Tweets | 97.81 | 1.51 |

Table 3: Number of distinct words and number of distinct time tokens in time expressions.

| Dataset | #Words | #Time Tokens |
|---------|--------|--------------|
| TimeBank | 130 | 64 |
| Gigaword | 214 | 80 |
| WikiWars | 224 | 74 |
| Tweets | 107 | 64 |

from 36.23% in WikiWars to 62.91% in Tweets. In informal communication people tend to use words in minimum length to express time information. The third column in Table 2 reports the average length of time expressions. On average, time expressions contain about two words.

**Observation 2** *Only a small group of time-related keywords are used to express time information.*

From the time expressions in all four datasets, we observe that the group of keywords used to express time information is small.

Table 3 reports the number of distinct words and of distinct time tokens. The words/tokens are manually normalized before counting and their variants are ignored. For example, 'month' and '5mons' are counted as one token 'month.' Numerals in the counting are ignored. Despite the different sizes in the four datasets, the numbers of distinct time tokens are comparable.

**Observation 3** *More than 93% of time expressions contain at least one time token.*

The second column in Table 2 reports the percentage of time expressions that contain at least one time token. Observe that at least 93.18% of time expressions contain time token(s), which suggests that to recognize the time expressions, it is essential to recognize their time tokens.

**Observation 4** *Part-of-speech (POS) could not distinguish time expressions from common words, but within time expressions, POS can help distinguish their constituents.*

For each dataset we list the top 10 POS tags that appear in time expressions, and their percentages over whole text. Among the 40 tags (10 × 4 datasets), 36 have percentage lower than 10%; other 4 are CD. This indicates that POS could not provide enough information to distinguish time expressions from common words. However, the most common POS tags in time expressions are NN*, JJ, RB, CD, and DT. Within time expressions, the time tokens usually have NN* and RB, the modifiers have JJ and RB, and the numerals have CD. This observation indicates that for the time expressions, their similar constituents behave in similar syntactic way. When seeing this, we realize that this is exactly how linguists define part-of-speech for language. [3] The definition of POS for language **inspires** us to define a syntactic type system for the time expression, part of language.

The four observations all relate to the principle of least effort (Zipf, 1949). That is, people will act under the least effort so as to minimize the cost of energy at both individual and collective level to the language usage (Zipf, 1949). Time expression is part of language and acts as an interface of communication. Short expressions, small vocabulary, occurrence, and similar syntactic behaviors all reduce the cost of energy required to communicate.

To summarize: On average, a time expression contains two tokens of which one is time token and the other is modifier/numeral, and the size of time tokens is small. To recognize a time expression, therefore, we first recognize the time token, then recognize the modifier/numeral.

# 4 SynTime: Syntactic Types and Simple Heuristic Rules

Figure 2 shows the overview of SynTime. Shown in the left-hand side of the figure, SynTime is initialized with regular expressions over tokens. After initialization SynTime can be directly applied on text, without training. On the other hand, SynTime can be easily expanded by adding time-related token regular expressions from training text. The expansion enables SynTime to recognize time expressions in text from different types and from different domains.

Shown in the right-hand side of Figure 2, SynTime recognizing time expression includes three main steps. In the first step, SynTime identifies

---

[3] "linguists group some words of language into classes (sets) which show similar syntactic behaviour." (Manning and Schutze, 1999)

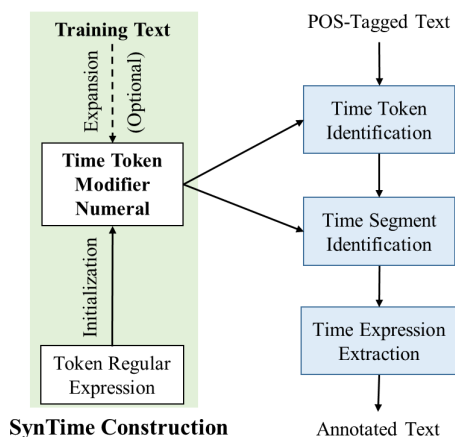

Figure 2: Overview of SynTime. Left-hand side shows the construction of SynTime, with initialization using token regular expressions, and optional expansion using training text. Right-hand side shows the main steps that SynTime recognizes time expressions.

the time tokens from POS-tagged raw text. Then around the time tokens SynTime searches for modifiers and numerals to form time segments. In the last step, SynTime transforms the time segments to time expressions.

## 4.1 SynTime Construction

We define **a syntactic type system** for time expression, specifically, 15 types for time tokens, 5 types for modifiers, and 1 type for numeral.

**Time Token.** We define 15 types for the time tokens and use their names similar to Joda-Time classes: [4] DECADE (-), YEAR (-), SEASON (5), MONTH (12), WEEK (7), DATE (-), TIME (-), DAY_TIME (27), TIMELINE (12), HOLIDAY (20), PERIOD (9), DURATION (-), TIME_UNIT (15), TIME_ZONE (6), and ERA (2). Number in '()' indicates the number of distinct tokens in this type, without counting variants. '-' indicates the type involves changing digits and cannot be counted.

**Modifier.** We define 3 types for the modifiers according to their possible positions relative to time tokens. Modifiers that appear before time tokens are PREFIX (48); modifiers after time tokens are SUFFIX (2). LINKAGE (4) link two time tokens. Besides, we define 2 special modifier types, COMMA (1) for comma ',' and IN_ARTICLE (2) for indefinite articles 'a' and 'an.'

---

[4] http://www.joda.org/joda-time/

TimeML (Pustejovsky et al., 2003a) and Time-Bank (Pustejovsky et al., 2003b) do not treat most prepositions like 'on' as a part of time expressions. Thus SynTime does not collect those prepositions.

**Numeral.** Number in time expressions can be a time token *e.g.,* '10' in 'October 10, 2016,' or a modifier *e.g.,* '10' in '10 days.' We define NUMERAL (-) for the ordinals and numbers.

**SynTime Initialization.** The token regular expressions for initializing SynTime are collected from SUTime,[5] a state-of-the-art rule-based tagger that achieved the highest recall in TempEval-3 (Chang and Manning, 2012, 2013). Specifically, we collect from SUTime **only** the tokens and the regular expressions over tokens, and **discard** its other rules of recognizing full time expressions.

## 4.2 Time Expression Recognition

On the types, SynTime designs **simple heuristic rules** to recognize time expressions. The recognition process includes three main steps: (1) Time token identification, (2) time segment identification, and (3) time expression extraction.

### 4.2.1 Time Token Identification

Identifying Time tokens is simple and straightforward, through **matching** of string and regular expression. Some words might cause ambiguity. For example 'May' could be a modal verb, or the fifth month of year. To filter out the ambiguous words, we use POS information. In implementation, we use Stanford POS Tagger, [6] and the POS tags for matching the instances of SynTime types are based on our Observation 4 in Section 3.2.

### 4.2.2 Time Segment Identification

The task of time segment identification is to search the surrounding of each time token identified in previous step for the modifiers and numerals, then gather the time token with its modifiers and numerals to form a time segment. The searching is under simple heuristic rules in which the key idea is to **expand the time token's boundaries**.

At first, each time token is a time segment. If it is either a PERIOD or DURATION, then no need to further search. Otherwise, search its left and its right for modifiers and numerals. For the left searching, if encounter a PREFIX or NUMERAL or IN_ARTICLE, then continue searching. For the right

---

[5] https://github.com/stanfordnlp/CoreNLP/tree/master/src/edu/stanford/nlp/time/rules
[6] http://nlp.stanford.edu/software/tagger.shtml

$$\overline{\phantom{xxx}s1\phantom{xxx}}\qquad\overline{\phantom{x}s2\phantom{x}}$$
the/PREFIX last/PREFIX week/TIME_UNIT … said Friday/WEEK
$$\underline{\phantom{xxx}e1\phantom{xxx}}\qquad\underline{\phantom{x}s1\phantom{x}}$$

(a) Stand-alone time segment to time expression.

$$\overline{\phantom{xxx}s1\phantom{xxx}}\qquad\overline{\phantom{x}s2\phantom{x}}$$
the/PREFIX third/NUMERAL quarter/TIME_UNIT of/PREFIXT 1984/YEAR
$$\underline{\phantom{xxxxxxxxxxx}s1\phantom{xxxxxxxxxxx}}$$

(b) Merge adjacent time segments.

$$\overline{\phantom{xx}s1\phantom{xx}}\quad\overline{\phantom{xx}s2\phantom{xx}}$$
January/MONTH 13/NUMERAL 1951/YEAR
$$\underline{\phantom{xxxxxxx}s1\phantom{xxxxxxx}}$$

(c) Merge overlapping time segments.

$$\overline{\phantom{xx}s1\phantom{xx}}\quad\overline{\phantom{xx}s2\phantom{xx}}$$
June/MONTH 30/NUMERAL ,/COMMA 1990/YEAR
$$\underline{\phantom{xxxxxxx}s1\phantom{xxxxxxx}}$$

(d) Merge overlapping time segments.

$$\overline{\phantom{xx}s1\phantom{xx}}\quad\overline{\phantom{xx}s2\phantom{xx}}$$
8/NUMERAL to/LINKAGE 20/NUMERAL days/TIME_UNIT
$$\underline{\phantom{x}e1\phantom{x}}\qquad\underline{\phantom{xx}s1\phantom{xx}}$$

(e) Dependent time segment and time segment.

Figure 3: Example time segments and time expressions. The labels above are from time segment identification; the labels below are for time expression extraction.

searching, if encounter a SUFFIX or NUMERAL, then continue searching. Both the left and the right searching will stop when reaching a COMMA or LINKAGE or non-modifier/numeral word. The left searching will not exceed the previous time token; the right searching will not exceed the next time token. A time segment consists of exactly one time token, and zero or more modifiers/numerals.

A special kind of time segments do not contain any time token; they depend on other time segments next to them. For example, in '8 to 20 days,' 'to 20 days' is a time segment, and '8 to' forms a dependent time segment. (see Figure 3(e).)

### 4.2.3 Time Expression Extraction

The task of time expression extraction is to extract time expressions from the identified time segments in which the core step is to determine whether to **merge** two adjacent or overlapping time segments into a new time segment.

We scan the time segments in a sentence from beginning to the end. A stand-alone time segment is a time expression. (see Figure 3(a).) The main focus is to deal with two or more time segments that are adjacent and overlapping. If two time segments $s_1$ and $s_2$ are adjacent, merge them to form a new time segment $s_1$. (see Figure 3(b).) Consider that $s_1$ and $s_2$ overlap at a shared boundary. According to our time segment identification, the shared boundary could be a modifier or a numeral. If the word at the shared boundary is neither a COMMA nor a LINKAGE, then merge $s_1$ and $s_2$. (see Figure 3(c).) If the word is a LINKAGE, then extract $s_1$ as a time expression and continue scanning. When the shared boundary is a COMMA, merge $s_1$ and $s_2$ only if the COMMA's previous token and its next token satisfy the three conditions: (1) The previous token is a time token or a NUMERAL; (2) the next token is a time token; and (3) the types of the previous token and of the next token are not the same. (see Figure 3(d).)

### 4.3 SynTime Expansion

SynTime expansion requires the words to be added to be annotated manually. We apply the initial SynTime on the time expressions from training text and list the words that are not covered. Whether the uncovered words are added to SynTime is manually determined. The rule for determination is that the added words can not cause ambiguity and should be generic. WikiWars dataset contains a few examples like this: 'The time Arnold reached Quebec City.' Words in this example are extremely descriptive, and we do not collect them. In tweets, on the other hand, people may use words' abbreviations and informal variants, for example, '2day' and 'tday' are popular spellings of 'today.' Such kind of abbreviations and informal variants will be collected.

According to our observations, not many words are used to express time information, the manual addition of keywords thus will not cost much. In addition, we find that even in tweets people tend to use formal words. In the Twitter word clusters trained from 56 million English tweets, [7] the most often used words are the formal words, and their frequencies are much greater than informal words'. The cluster of 'today,' [8] for example, its most often use is the formal one, 'today,' which appears 1,220,829 times; while its second most often use '2day' appears only 34,827 times. The low rate of informal words (*e.g.,* about 3% in 'today' cluster) suggests that even in informal environment the manual keyword addition costs little.

---

[7] http://www.cs.cmu.edu/~ark/TweetNLP/cluster_viewer.html

[8] http://www.cs.cmu.edu/~ark/TweetNLP/paths/01111110010.html

## 5 Experiments

We conduct experiments on three datasets and compare SynTime with three state-of-the-art baselines: HeidelTime, SUTime, and UWTime. For SynTime we report the results of its two versions: **SynTime-I** and **SynTime-E**. SynTime-I is the initial version, and SynTime-E is the expanded version by adding keywords to SynTime-I.

### 5.1 Experiment Setting

**Datasets.** We use two benchmark datasets, Time-Bank and WikiWars, and one manually labeled Tweets dataset. Section 3.1 shows details of Time-Bank and WikiWars datasets. The Tweets dataset is collected from Twitter. We randomly sample 4000 tweets and use SUTime to tag them. 942 tweets of which each contains at least one time expression. From the remaining 3,058 tweets, we randomly sample 500 and manually annotate them, and find that only 15 tweets contain time expressions. Thus we roughly consider that SUTime misses about 3% time expressions in tweets. We then manually annotate the 942 tweets, according to the standards of TimeML and TimeBank, and get 1,127 manually labeled time expressions. For the 942 tweets, we randomly sample 200 tweets as test set, and the rest 742 as training set, because a baseline UWTime requires training.

**Baseline Methods.** We compare SynTime with methods: HeidelTime (Strötgen and Gertz, 2010), SUTime (Chang and Manning, 2012), and UW-Time (Lee et al., 2014). HeidelTime and SU-Time both are rule-based methods, and UWTime is a learning method. When training UWTime on Tweets, we try two settings: (1) Train with only Tweets training set; (2) train with TimeBank and Tweets training set. The second setting achieves slightly better result and we report that result.

**Evaluation Metrics.** We follow TempEval-3 and use their evaluation toolkit [9] to report $Precision$, $Recall$, and $F_1$ in terms of *strict match* and *relaxed match* (UzZaman et al., 2013).

### 5.2 Experiment Result

Table 4 reports the overall performance. Among the 18 measures, SynTime-I and SynTime-E achieve 12 best results and 13 second best results. Except the strict match on WikiWars dataset,

---

[9] http://www.cs.rochester.edu/~naushad/tempeval3/tools.zip

---

both SynTime-I and SynTime-E achieve $F_1$ above 91%. For the relaxed match on all three datasets, SynTime-I and SynTime-E achieve recalls above 92%. The high recalls are consistent with our observation that at least 93.18% of time expressions contain time token(s). (see Table 2.) This indicates that SynTime covers most of time tokens. On Tweets dataset, SynTime-I and SynTime-E achieve exceptionally good performance. Their $F_1$ reach 91.74% with 11.37% improvement in strict match and 95.87% with 6.33% improvement in relaxed match. The reasons are that in informal environment people tend to use time expressions in minimum length, (62.91% of one-word time expressions in Tweets; see Figure 1.) the size of time keywords is small, (only 60 distinct time tokens; see Table 3.) and even in tweets people tend to use formal words. (see Section 4.3 for our finding from Twitter word clusters.) For precision, Syn-Time achieves comparable results in strict match and performs slightly poorer in relaxed match.

#### 5.2.1 SynTime-I v.s. Baseline Methods

On TimeBank dataset, SynTime-I achieves $F_1$ of 92.09% in strict match and of 94.96% in relaxed match. On Tweets, SynTime-I achieves 91.74% and 95.87%, respectively. It outperforms all the baseline methods. The reason is that for the rule-based time taggers, their rules are designed in a *fixed* way, lacking flexibility. For example, SU-Time could recognize '1 year' but not 'year 1.' For the machine learning based methods, some of the features they used actually hurt the modelling. Time expressions involve quite many changing numbers which in themselves affect the pattern recognition. For example, it is difficult to build connection between 'June 30, 1990' and 'October 12, 2008' at the level of word or of character. One suggestion is to consider a type-based learning method that could use type information. For example, the above two time expressions refer to the same pattern of 'MONTH NUMERAL COMMA YEAR' at the level of type. POS is a kind of type information. But according to our analysis, POS could not distinguish time expressions from common words. Features need carefully designing. On WikiWars, SynTime-I achieves competitive results in both matches. Time expressions in Wiki-Wars include lots of prepositions and quite a few descriptive time expressions. SynTime could not fully recognize such kinds of time expressions because it follows TimeML and TimeBank.

Table 4: Overall performance. The **best results** are in bold face and the second best are underlined. Some results are borrowed from their original papers and the papers are indicated by the references.

| Dataset | Method | Strict Match | | | Relaxed Match | | |
|---|---|---|---|---|---|---|---|
| | | $Pr.$ | $Re.$ | $F_1$ | $Pr.$ | $Re.$ | $F_1$ |
| TimeBank | HeidelTime(Strotgen et al., 2013) | 83.85 | 78.99 | 81.34 | 93.08 | 87.68 | 90.30 |
| | SUTime(Chang and Manning, 2013) | 78.72 | 80.43 | 79.57 | 89.36 | 91.30 | 90.32 |
| | UWTime(Lee et al., 2014) | 86.10 | 80.40 | 83.10 | **94.60** | 88.40 | 91.40 |
| | SynTime-I | 91.43 | 92.75 | 92.09 | 94.29 | **95.65** | **94.96** |
| | SynTime-E | **91.49** | **93.48** | **92.47** | 93.62 | **95.65** | 94.62 |
| WikiWars | HeidelTime(Lee et al., 2014) | 85.20 | 79.30 | 82.10 | 92.60 | 86.20 | 89.30 |
| | SUTime | 78.61 | 76.69 | 76.64 | 95.74 | 89.57 | 92.55 |
| | UWTime(Lee et al., 2014) | **87.70** | 78.80 | **83.00** | **97.60** | 87.60 | 92.30 |
| | SynTime-I | 80.00 | 80.22 | 80.11 | 92.16 | 92.41 | 92.29 |
| | SynTime-E | 79.18 | **83.47** | 81.27 | 90.49 | **95.39** | **92.88** |
| Tweets | HeidelTime | **89.58** | 72.88 | 80.37 | 95.83 | 77.97 | 85.98 |
| | SUTime | 76.03 | 77.97 | 76.99 | 88.43 | 90.68 | 89.54 |
| | UWTime | 88.54 | 72.03 | 79.44 | **96.88** | 78.81 | 86.92 |
| | SynTime-I | 89.52 | 94.07 | 91.74 | 93.55 | 98.31 | 95.87 |
| | SynTime-E | 89.20 | **94.49** | **91.77** | 93.20 | **98.78** | **95.88** |

Table 5: Number of time tokens and modifiers for expansion

| Dataset | #Time Tokens | #Modifiers |
|---|---|---|
| TimeBank | 4 | 5 |
| WikiWars | 16 | 19 |
| Tweets | 3 | 2 |

### 5.2.2 SynTime-E v.s. SynTime-I

Table 5 lists the number of time tokens and modifiers added to SynTime-I to get SynTime-E.

On TimeBank and Tweets datasets, only a few tokens are added, the corresponding results are affected slightly. This confirms the small size of time words and the high coverage of SynTime. On WikiWars, relatively more tokens are added, SynTime-E performs much better than SynTime-I, especially in recall. It improves the recall by 3.25% in strict match and by 2.98% in relaxed match. This indicates that with more words added from specific domains (*e.g.,* WikiWars about war), SynTime can improve the performance.

### 5.3 Limitations

SynTime assumes that words are tokenized and POS tagged correctly. In reality, however, the tokenized and tagged words are not that perfect, due to the limit of used tools. For example, Stanford POS Tagger assigns VBD to the word 'sat' in 'friday or sat' while whose tag should be NNP. The incorrect tokens and POS tags affect the result.

## 6 Conclusion and future work

We conduct an analysis on the time expressions from four datasets, and observe that time expressions in general are very short and expressed by a small vocabulary, and words in time expressions demonstrate similar syntactic behavior. Our **observations** provide evidence in terms of time expression for **the principle of least effort** (Zipf, 1949). Inspired by part-of-speech, based on the observations, we define a syntactic type system for the time expression, and propose a type-based time expression tagger, named by SynTime. SynTime defines **syntactic types** for tokens and on the types it designs **simple heuristic rules** based on the idea of **boundary expansion**. Experiments on three datasets show that SynTime outperforms the state-of-the-art baselines, including rule-based time taggers and machine learning based time tagger. As an extremely light-weight rule-based tagger, SynTime runs in real time. SynTime provides an idea to **simplify** the rule-based time tagger.

Time expression is part of language and follows the principle of least effort (Zipf, 1949). Since language usage relates to human habits (Chomsky, 1986; Pinker, 1995), we might expect that humans would share some common habits, and therefore expect that other languages and other parts of language would more or less follow the same principle. In the future we will try our analytical method on other languages and other parts of language.

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
