# Peer review of "Time Expression Analysis and Recognition Using Syntactic Token Types and General Heuristic Rules"

_ACL 2017 — decision unknown_

[Official Review · Reviewer 1 · rating 4 · confidence 3]
soundness 4 · originality 3 · clarity 4 · substance 4 · appropriateness 5 · presentation format Oral Presentation

This paper describes a rule based approach to time expression extraction. Its
key insights are time expressions typically are short and contain at least 1
time token. It first recognizes the time token through a combination of
dictionary lookup, regular expression match with POS tagging information. It
then expands the time segment from either direction of the time token until it
reaches based on a set of heuristic rules. Finally, it merges the time segments
into a single time expression based on another set of rules. Evaluation of this
approach with both rule based & ML based systems on 3 data sets show
significant improvements.

- Strengths:

It's well written and clearly presented. The rules are motivated by empirical
observations of the data, and seems to be well justified as evidenced by the
evaluation. 

- Weaknesses:

 There are some underspecification in the paper that makes it difficult to
reproduce the results. See below for details.

- General Discussion:

* Section 4.1: what are there 5 seasons? What about things such as Ramadan
month or Holiday Season?
* Section 5.1: "two benchmark datasets" => "three datasets"?
* Section 5.2: an example without time token will be helpful.
* Section 5.2: given this approach is close to the ceiling of performance since
93% expressions contain time token, and the system has achieved 92% recall, how
do you plan to improve further?
* Is there any plan to release the full set of rules/software used?

[Official Review · Reviewer 2 · rating 3 · confidence 4]
soundness 4 · originality 3 · clarity 4 · substance 4 · appropriateness 4 · presentation format Poster

The paper proposes a method to recognize time expressions from text. It is a
simple rule-based method, which is a strong advantage as an analysis tool since
time expression recognition should be a basic process in applications.
Experiments results show that the proposed method outperforms the
state-of-the-art rule-based methods and machine learning based method for time
expression recognition. 

It is great, but my concern is generality of the method. The rules in the
method were designed based on observations of corpora that are used for
evaluation as well. Hence I’m afraid that the rules over-fit to these
corpora. Similarly, domains of these corpora may have affected the rule design.
There is no statistic nor discussion to show overlaps in time expressions in
the observed corpora. If it was shown that time expressions in these corpora
are mostly overlap, the fact should have supported generality of the rules. 

Anyway, it was better that the experiments have been conducted using a new
corpus that was distinct from rule design process in order to show that the
proposed method is widely effective.